# RecombineX: A generalized computational framework for automatic high-throughput gamete genotyping and tetrad-based recombination analysis

Jing Li[1,2], Bertrand Llorente[3], Gianni Liti[2]*, Jia-Xing Yue[1,2]*

**1** State Key Laboratory of Oncology in South China, Collaborative Innovation Center for Cancer Medicine, Guangdong Key Laboratory of Nasopharyngeal Carcinoma Diagnosis and Therapy, Sun Yat-sen University Cancer Center, Guangzhou, China, **2** Université Côte d'Azur, CNRS, INSERM, IRCAN, Nice, France, **3** Aix-Marseille Université, CNRS, INSERM, CRCM, Institut Paoli-Calmettes, Marseille, France

* gianni.liti@unice.fr (GL); yuejiaxing@gmail.com (JXY)

**Data Availability Statement:** The RecombineX software is freely distributed under MIT license at GitHub (https://github.com/yjx1217/RecombineX). The Oxford Nanopore reads as well as the

## Abstract

Meiotic recombination is an essential biological process that ensures faithful chromosome segregation and promotes parental allele shuffling. Tetrad analysis is a powerful approach to quantify the genetic makeups and recombination landscapes of meiotic products. Here we present RecombineX (https://github.com/yjx1217/RecombineX), a generalized computational framework that automates the full workflow of marker identification, gamete genotyping, and tetrad-based recombination profiling based on any organism or genetic background with batch processing capability. Aside from conventional reference-based analysis, RecombineX can also perform analysis based on parental genome assemblies, which facilitates analyzing meiotic recombination landscapes in their native genomic contexts. Additional features such as copy number variation profiling and missing genotype inference further enhance downstream analysis. RecombineX also includes a dedicate module for simulating the genomes and reads of recombinant tetrads, which enables fine-tuned simulation-based hypothesis testing. This simulation module revealed the power and accuracy of RecombineX even when analyzing tetrads with very low sequencing depths (e.g., 1-2X). Tetrad sequencing data from the budding yeast *Saccharomyces cerevisiae* and green alga *Chlamydomonas reinhardtii* were further used to demonstrate the accuracy and robustness of RecombineX for organisms with both small and large genomes, manifesting RecombineX as an all-around one stop solution for future tetrad analysis. Interestingly, our re-analysis of the budding yeast tetrad sequencing data with RecombineX and Oxford Nanopore sequencing revealed two unusual structural rearrangement events that were not noticed before, which exemplify the occasional genome instability triggered by meiosis.

## Author summary

Meiosis is a fundamental cellular process that ensures faithful chromosome segregation and promotes allele shuffling. Tetrad analysis, which isolates and genotypes all four

corresponding genome assemblies of the gametes AND1702-8:a and AND1702-12:a have been deposited to the SRA database under the accession number of PRJNA698424. In addition, another copy of the genome assembly of AND1702-8:a and AND1702-12:a together with the corresponding genome annotations have been uploaded to RecombineX's GitHub depository under the data subfolder (https://github.com/yjx1217/RecombineX/tree/master/data). The key raw data are uploaded to the Research Deposit public platform (www.researchdata.org.cn), with the approval RDD number of RDDB2021509656. The underlying numerical data for all the graphs presented in the manuscript is in the Supporting Information.

**Funding:** This work is supported by National Natural Science Foundation of China (32070592 to J.-X. Y. and 32000395 to J. L.), Guangdong Basic and Applied Basic Research Foundation (2019A1515110762 to J.-X. Y), Guangdong Pearl River Talents Program (2019QN01Y183 to J.-X. Y), Microsoft Azure Research Award (CRM:074871 to J.-X. Y.), ANR (ANR-15-IDEX-01 and ANR-20-CE13-0010 to G. L.; ANR-18-CE12-0013 to B. L.), Fondation pour la Recherche Médicale (EQU202003010413 to G. L), and Guangzhou Municipal Science and Technology Bureau (202102020938 to J. L.), respectively. The funders have not played any role in the study design, data collection and analysis, decision to publish, or preparation of the manuscript.

**Competing interests:** The authors have declared that no competing interests exist.

meiotic products (i.e., tetrad) derived from a single meiosis, remains the most straightforward and powerful way of studying meiotic recombination and its modulators at fine scales. The wide application of tetrad analysis in yeasts, filamentous fungi, green algae, and land plants have substantially expanded our understanding of meiotic recombination in terms of both genome-wide landscapes and molecular mechanisms. Here we described the first generalized computational framework named RecombineX that automates the full workflow of tetrad analysis based on any organism or genetic background. In addition, aside from conventional reference-based analysis, RecombineX can also perform analysis based on parental genome assemblies, which enables analyzing meiotic recombination landscapes in their native genomic contexts. Using both simulated and real tetrad-sequencing data, we further demonstrated RecombineX's trustable performance, versatile usage, and batch-processing capability, manifesting RecombineX as an all-around one stop solution for tetrad analysis. Especially considering that meiotic gamete genome sequencing from different natural and mutant backgrounds can now be acquired, we expect RecombineX to become a popular tool that empowers future tetrad analysis across different genetic backgrounds and species.

## Introduction

Meiosis is a fundamental cellular process in eukaryotes, through which sexually reproducing organisms generate their gametes typically via two successive rounds of cell division. In the first round (meiosis I), homologous chromosomes duplicate, pair and swap genetic materials, and then segregate into two daughter cells. In the next round (meiosis II), the two sets of sister chromatids in each daughter cell further separate into different gamete cells to reduce the total chromosome number by half. The four gamete cells resulting from these two rounds of cell division are collectively referred as a tetrad. In most species, accurate homologous chromosome segregation at meiosis I relies on sister chromatid cohesion in combination with meiotic crossovers (CO) that are reciprocal exchanges of chromosome arms. These meiotic COs result from the repair by homologous recombination of meiotic prophase-induced DNA doubles strand breaks (DSBs). In addition to COs, DSB repair by meiotic recombination also produces recombinants without reciprocal exchange of chromosome arms called non-crossovers (NCOs). Both COs and NCOs are intrinsically associated with a tract of gene conversion (GC), whose detection relies on suitably positioned markers. Both COs and NCOs shuffle parental genetic materials, which promotes the genetic robustness and phenotypic potential of the offspring gene pool.

Given the vital role of meiotic recombination, different methodologies have been developed to characterize its underlying mechanisms and evolutionary implications. For example, meiotic recombination can be indirectly analyzed by examining linkage disequilibrium and haplotype structure with population genomics data [1–4]. While powerful statistical inferences regarding recombination can be made in this way with existing genomic data, additional factors such as demographic histories and selection schemes might perplex the result interpretation. In contrast, meiotic recombination can also be studied by directly examining the makeup of gamete genotypes in terms of parental genetic backgrounds. One way of doing this is to perform bulk genotyping analysis for random gametes [5–8]. Although this approach allows for detailed delineation of cumulative recombination landscapes across a large number of gametes, it lacks the power and resolution for analyzing individual meioses, which prevents an in-depth view of the meiotic recombination process. Alternatively, at least for a selection of

model systems, it is feasible to isolate and genotype all four meiotic products (i.e., tetrad) derived from a single meiosis. This approach is called "tetrad analysis", which remains the most straightforward and powerful way of studying meiotic recombination and its modulators at fine scales. For instance, a landmark study of this kind was performed on the budding yeast *Saccharomyces cerevisiae*, which led to the first high-resolution meiotic recombination map for eukaryotes [9]. Thereafter, similar tetrad-based genome analysis have been carried out across multiple organisms and genetic backgrounds (including mutants) [10–18], which altogether substantially advanced our understanding of meiotic recombination and its genetic modulators.

In contrast to the broad application of tetrad analysis, there is a lack of dedicated computational framework for corresponding data analysis. To our knowledge, ReCombine [19] is the only tool developed for such purpose so far. ReCombine represents an important step towards automated and standardized tetrad analysis, but it was designed in the early days of next-generation sequencing and understandably appears somewhat constrained to cover today's use scenarios in terms of functionality, versatility, and customizability. For example, ReCombine is hardcoded based on the *S. cerevisiae* reference genome and expects the reference-based S288C strain to be one of the two crossing parents. Also, manual configuration and curation are normally needed on a tetrad-by-tetrad basis, making ReCombine less suitable for processing large numbers of tetrads.

Therefore, a new generation of computational solution for high-throughput and high-quality tetrad analysis is much needed. Here we introduce RecombineX, a generalized computational framework that automates the full workflow of gamete sequencing data analysis, especially for organisms whose tetrad can be isolated. Equipped with dedicated modules for polymorphic marker identification, gamete genotyping, recombination profiling, and recombinant tetrad simulation, RecombineX shines in its trustable performance, versatile usage, and batch-processing capability. Our tests based on both simulated and real data further demonstrated its consistent power and accuracy in multiple application scenarios, manifesting RecombineX as an all-around one stop solution for future tetrad analyses.

## Results and discussion

### The general design of RecombineX

RecombineX is a Linux-based computational framework designed for automated high-throughput tetrad analysis. It is self-contained by design and can be automatically installed and configured via a pre-shipped installer script. RecombineX comes with a series of task-specific modules handling different workflow phases: genome and read preparation -> parental marker identification -> gamete read mapping and genotyping -> tetrad-based recombination event profiling (Fig 1). Depending on the available input data, RecombineX can be executed in two modes: 1) the reference-based mode and 2) the parent-based mode. The reference-based mode requires a reference genome assembly as well as the short reads of the two crossing parents. The parent-based mode requires the native genome assemblies of both crossing parents, while the short reads of the two crossing parents can be further used when available (S1 Fig). Both reference-based mode and parent-based mode further need short reads of individual gametes, ideally labeled based on their original tetrad contexts. For each mode, we numbered the corresponding modules based on their execution orders, ensuring a well-organized data analysis workflow. Within each module, a task-specific executable bash script is provided for invoking the corresponding module. A template directory with these modules pre-configured is further provided as a testing example, which can be easily adapted for users' own project (S2 Fig).

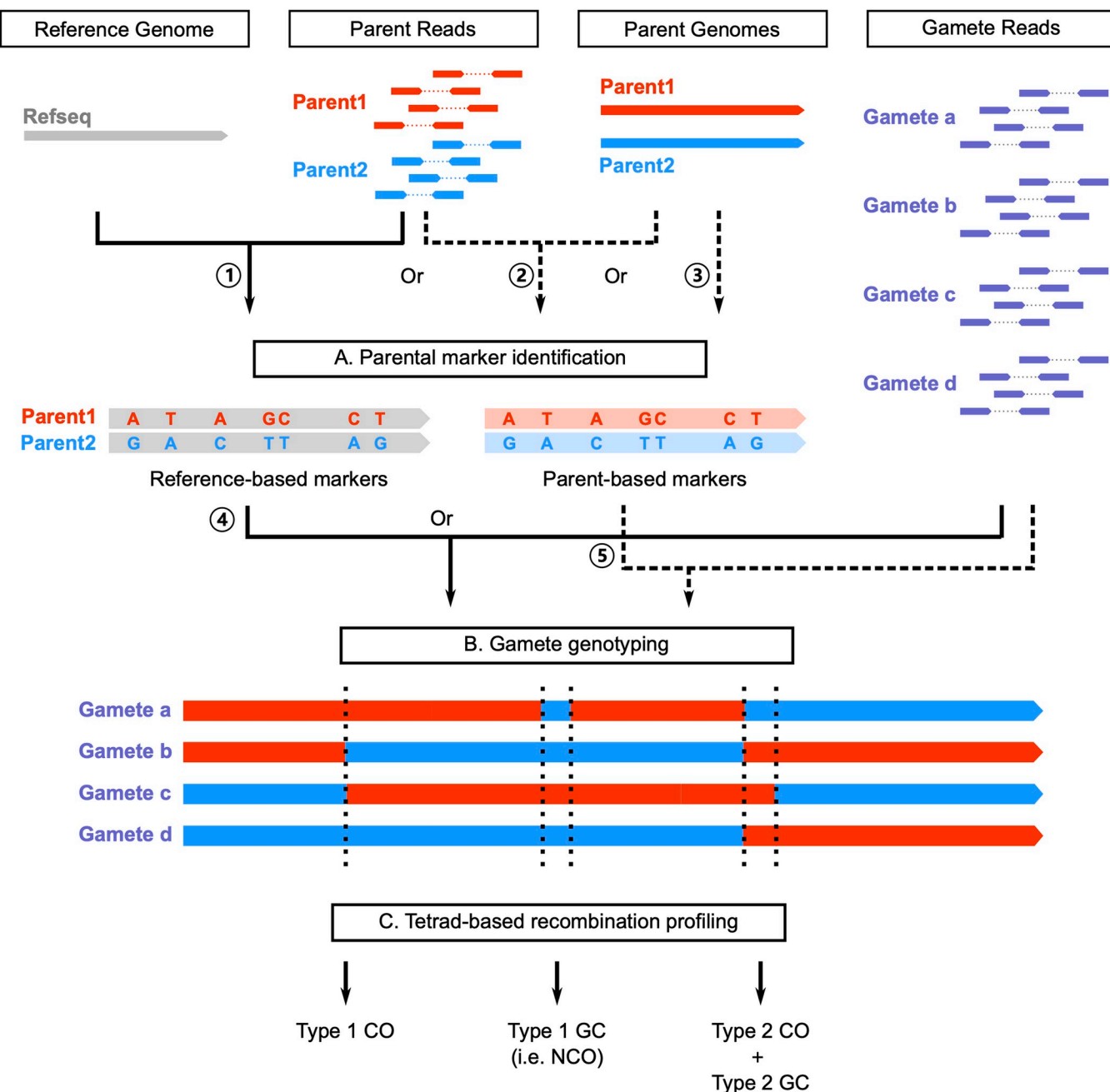

**Fig 1. An overview of the RecombineX framework.** RecombineX conduct sequencing-based tetrad analysis in three major phases: A) parental marker identification, B) gamete genotyping, and C) tetrad-based recombination profiling. Depending on the available input data, users can run RecombineX in either reference-based mode (denoted by solid arrows) or parent-based mode (denoted by dashed arrows). In the reference-based mode, parent reads are mapped to the reference genome for reference-based parental marker identification (①), based on which gamete genotyping is further performed by evaluating the gamete-to-reference read mapping support at each marker position (④). The resulting genotyping assignments across the four gametes from the same tetrad are jointly evaluated for profiling recombination events based on the reference genome coordinate system. In the parent-based mode, whole genome alignment is firstly constructed based on the native genome assemblies of the two crossing parents, upon which parent-based markers are identified accordingly (③). Optionally, parent-based markers obtained from whole genome alignment can be further leveraged by reciprocal parent-based read mapping (②). In either case, gamete genotyping is performed by evaluating the gamete-to-parent read-mapping at each marker position (⑤). The resulting genotyping assignments across the four gametes from the same tetrad are jointly evaluated for profiling recombination events based on the coordinate systems of the two parental genome assemblies.

In the reference-based mode, the sequencing reads of the two crossing parents are mapped to the reference genome to identify single nucleotide variants (SNVs) between the two parental backgrounds (S3 Fig). High-confident SNVs that are free from repetitive sequences and copy number variants (CNVs) are used as polymorphic markers segregated between the two parental backgrounds. Both haploid and diploid parent genomes are supported, and in both cases only homozygous SNVs will be considered. Such design choice helps avoiding the risk of including ambiguous markers for most use scenarios, although understandably limits the application of RecombineX on diploid parental genomes with very high heterozygosity. After parental marker identification, gamete reads are subsequently mapped to the reference genome, upon which RecombineX computes the best supported genotype at each marker position for each gamete (S4 Fig). A genotype purity filter is further employed at this step to cull out markers with clear admixed genotype signals. Such admixed genotype signals normally come from ambiguous mapping and therefore it is reasonable to filter them out in normal tetrad analysis. However, admixed genotype signals can also reveal post-meiotic segregation (PMS) of unrepaired heteroduplex DNA (mismatches) formed during recombination whose frequency massively increases after inactivation of the mismatch repair machinery. PMS is important for dissecting the detailed molecular mechanisms of recombination. PMS is directly detectable in filamentous fungi that naturally form octads or after micromanipulation of yeast tetrads [11,17,20,21]. To allow PMS identification, we added an option that enables RecombineX to report all marker sites with admixed genotype signals. Also, when needed, RecombineX can infer marker-specific missing genotypes by assuming a tetrad-wide 2:2 segregation ratio between the two parental backgrounds, which could come handy to recover genotypes that are otherwise inaccessible. By jointly analyzing genotype segregation and switching patterns within the same tetrad across different marker positions, RecombineX further identifies and classifies recombination events into different categories. In general, profiled recombination events are expected to fall in four major categories: CO without associated GC (referred as Type 1 CO thereafter), NCO (referred as Type 1 GC thereafter), CO with associated GC (referred as Type 2 CO thereafter), and the Type 2 CO associated GC (referred as Type 2 GC thereafter), although more complex cases could be encountered occasionally (S5 and S6 Figs). It is worth pointing out that a CO event always associates with a GC tract and therefore Type 1 and Type 2 CO events are biologically equivalent. It is due to lacking available markers that makes the CO-associated GC tracts undetectable in practice. For all profiled recombination events, RecombineX generates detailed reports on their genomic coordinates, marker supports, genotype segregation patterns, and the associated linkage blocks for downstream analysis.

In addition to the reference-based tetrad analysis described above, RecombineX also supports performing tetrad analysis directly based on the genome assemblies of the two crossing parents, which could be especially valuable for analyzing recombination events in their native parental genome contexts. In this parent-based mode, RecombineX builds the whole genome alignment of the two parental genome assemblies and identifies parental markers accordingly (S3 Fig). When available, sequencing reads of the two parents can be further provided to derive consensus markers by further leveraging cross-parent read mapping results (S3 Fig). Again, heterozygous SNVs or SNVs located in repetitive or CNV regions will be filtered out during marker identification. Upon the identification of parental markers, the downstream analyses such as gametes reads mapping, gametes genotyping, and tetrad-based recombination profiling are performed based on both parental genome coordinate systems (S4 Fig), with the corresponding results also reported in two mirrored copies. In this way, users can easily check for association between identified recombination events and various parental genome features (e.g., gene densities, repetitive sequence abundance, GC% contents, parental divergence, DSB hotspots, etc.) based on the same genome coordinate system.

## Simulation-based validation for RecombineX's parental marker identification modules

Accurate and robust parental markers identification is a prerequisite for high-quality tetrad analysis. Several studies demonstrated that downstream recombination analysis can be severely compromised when relying on ambiguous markers, which are often derived from genomic regions associated with repetitive sequences or CNVs [13,22]. Therefore, with RecombineX, we designed and implemented multiple filters to effectively cull out markers falling in repetitive and CNV regions (S3 Fig; See Materials and Methods for details). As a simulation-based test, we let RecombineX to identify markers segregated between two hypothetical parental genomes: the *S. cerevisiae* reference genome and a simulated *S. cerevisiae* genome with 60,000 SNVs, 6,000 INDELs, and 6 CNVs, mimicking a typical 0.5% genomic divergence between two *S. cerevisiae* natural isolates [23]. Among the 60,000 simulated SNV sites, 8,087 sites fell in either repetitive or CNV regions, leaving the remaining 51,913 sites as valid targets for marker identification (Fig 2 and S1 Table). Based on these two hypothetical parental genomes, we evaluated RecombineX's marker identification performance in both reference-based and parent-based modes. In reference-based mode, RecombineX solely relies on the input parent reads to identify markers. While the power of marker identification positively correlates with parent sequencing depth, it quickly enters diminishing return with sequencing depth >30X (Fig 2 and S1 Table). Therefore, as a rule of thumb, we generally recommend using > = 30X parental reads for marker identification with RecombineX. Based on our

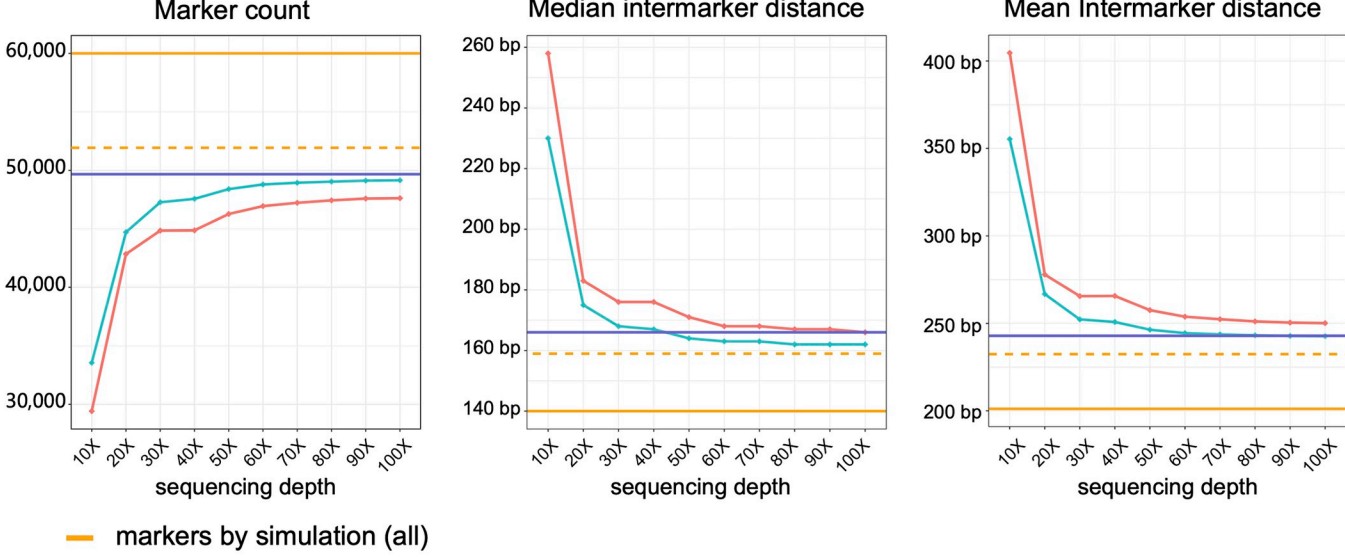

**Fig 2. Performance of polymorphic marker identification with RecombineX.** A total of 60,000 SNV markers (denoted as "all"; solid orange line) together with 6,000 Indels and 6 CNVs were simulated for two hypothetical parent genomes, among which 51,913 of them are considered as valid marker discovery targets (denoted as "valid"; dashed orange line) as they are not associated with repetitive regions nor CNV regions. We gauged RecombineX's performance for marker identification based on these two hypothetical parent genomes and their reads (simulated sequencing coverage: 10X, 20X, 30X, . . ., 100X) using different marker identification protocols implemented in RecombineX: reference-based mapping (solid green line), whole genome alignment (solid purple line), and consensus between whole genome alignment and parent-based read mapping (solid red line). For each case, the total marker counts as well as the median and mean inter-marker distance were plotted respectively.

simulation, RecombineX is able to recover >90% valid marker targets (47,268 out of 51,913) with 30X parent reads. A close examination of these markers shows no false positive calling was made and ambiguous SNV sites from repeat-/CNV-associated regions have been effectively filtered out, proving the high reliability of RecombineX's marker identification. The only CNV-associated SNV site that escaped from RecombineX's CNV-filter locates near the boundary of a simulated CNV with no detectable per-site mapping depth deviation from the chromosome-wide median. In parent-based mode, RecombineX identified 49,669 markers based on parental genome alignment alone (referred as markers identified by whole-genome-alignment). When parental reads were further supplied for reciprocal parent read mapping and consensus marker identification, the corresponding consensus marker count scales with the sequencing depth of parent reads. Notably, the improvement of the consensus marker count becomes marginal with sequencing depth > 50X. Although RecombineX has implemented a unique-alignment-based CNV filter for genome-alignment-based marker identification, we found 38 out of 8,087 CNV-associated markers escaped from this filter. Nearly all of them (37 out of 38) were further filtered out when calling consensus markers, during which an additional mapping-depth-based CNV filter is applied. Therefore, the consensus marker identification protocol appears more robust against ambiguous markers from CNV regions. The only marker that escaped from both CNV filters is the same one as mentioned above, which shows very weak CNV signal in both reference-based and parent-based modes. According to this result, we recommend opting for consensus marker identification strategy when running RecombineX in parent-based mode, if parent reads are available.

## Simulation-based validation for RecombineX's gamete genotyping module

To assess RecombineX's gamete genotyping accuracy, we used RecombineX's built-in simulation module to simulate one recombinant tetrad derived from two hypothetical crossing parents: P1 and P2. The inputs of this simulation module include a reference or parent genome to set up the coordinate system and a list of parental markers projected to the same coordinate system. Here we used the *S. cerevisiae* reference genome as the coordinate system and the above identified 49,153 reference-based markers based on 100X parent reads (Fig 2 and S1 Table). For the simulated recombinant tetrads, we introduced CO and NCO events based on the count and size parameters estimated from real yeast tetrads [9] (Fig 3A). For the four resulting recombinant gametes, sequencing reads of varied depths (1X, 2X, 4X, 8X, 16X, 32X, 64X) were further simulated. With this simulated dataset, we evaluated RecombineX's genotyping performance for mimicked scenarios of both fully and partially viable tetrads. For the scenario of fully viable tetrad, the simulated reads of all four gametes were used for genotyping (Fig 3B). For the scenario of a partially viable tetrad, the simulated reads of only 3 gametes were used, leaving the remaining one to be inferred by RecombineX (Fig 3C and 3D). RecombineX infers missing genotypes by assuming a tetrad-wide 2:2 segregation ratio between the two parental genotypes across the whole genome. While such ratio can deviate from 2:2 in genomic regions with GC tracts, the cumulative size of all GC tracts is typically orders of magnitude smaller in comparison to the genome size, making this assumption hold in general. Finally, it is worth emphasizing that although a pre-specified gamete-tetrad correspondence enables extra features such as missing genotyping inference, RecombineX's raw genotyping function does not require any tetrad information as the *priori* when performing raw genotyping. Therefore, users can also use RecombineX to perform plain genotyping analysis for random gametes derived from known parents. Moreover, additional tools have been previously developed for reconstructing the gamete-tetrad correspondence map from random gametes

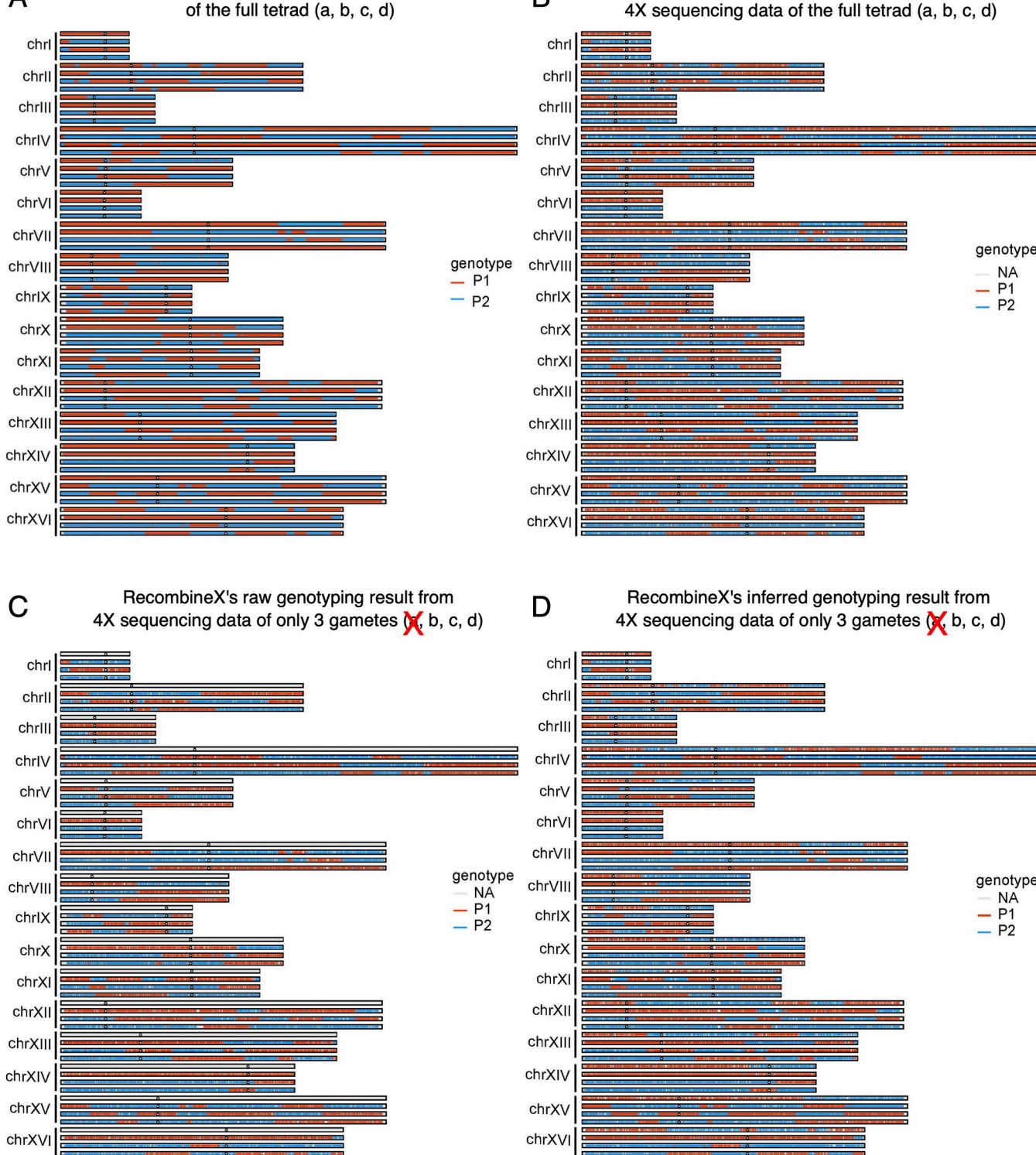

**Fig 3. Performance of raw and inferred gamete genotyping with RecombineX.** (A) The ground truth genotypes of the simulated tetrad (a, b, c, d). (B) RecombineX's raw genotyping result for the simulated tetrad based on 4X sequencing data with a $Q_{net}$ cutoff of 30. (C) The raw genotyping result for the simulated tetrad based on 4X sequencing data of only three gametes (b, c, d) with a $Q_{net}$ cutoff of 30. (D) The inferred genotyping result for the simulated tetrad based on 4X sequencing data of only three gametes (b, c, d) with a $Q_{net}$ cutoff of 30.

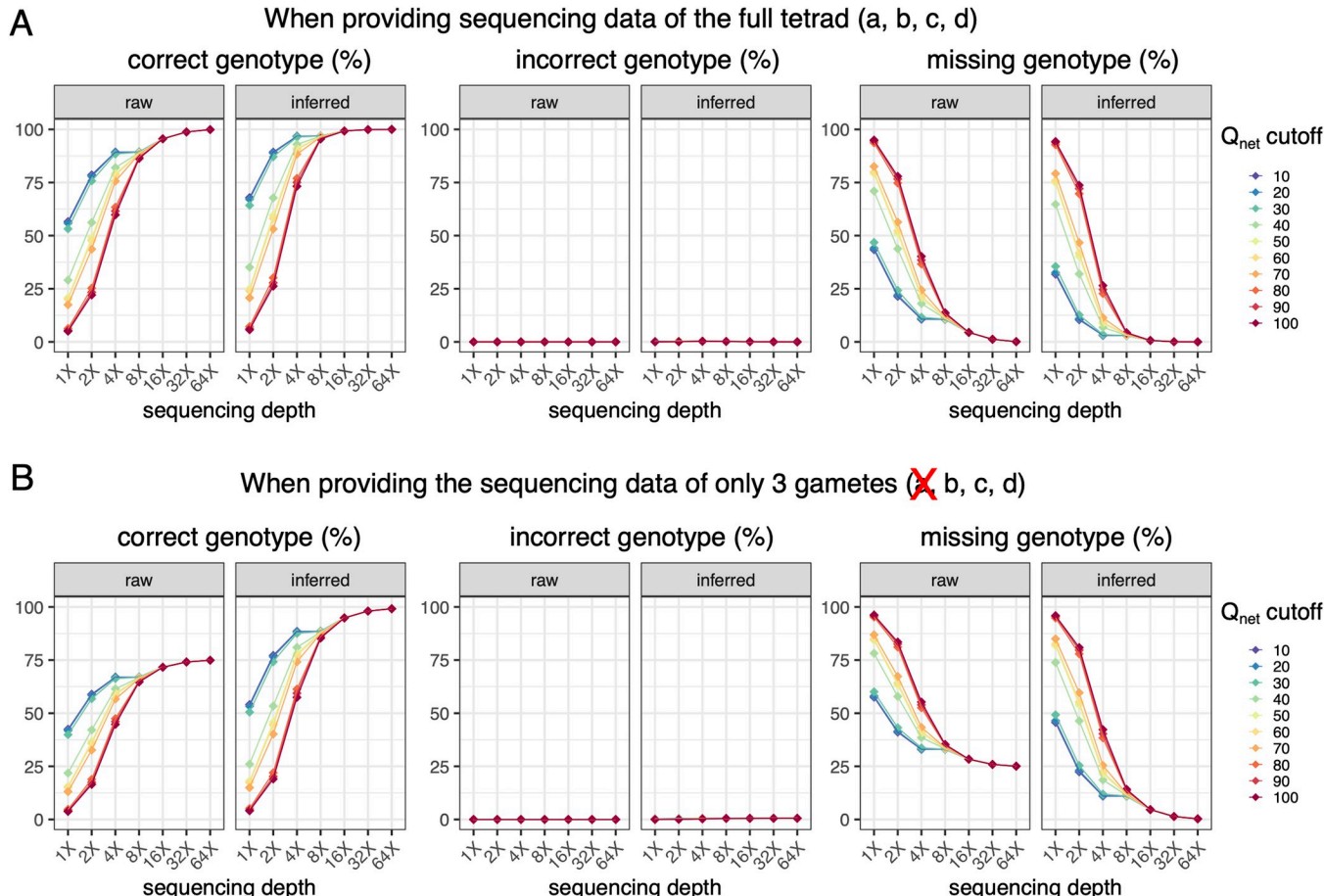

**Fig 4. The benchmark of RecombineX's genotyping accuracy.** (A) The calculated percentage of correct, incorrect, and missing genotypes based on RecombineX's gamete genotyping results in comparison to the simulated ground truth when the sequencing data of all four gametes (a, b, c, d) are provided. (B) The calculated percentage of correct, incorrect, and missing genotypes based on RecombineX's genotyping results in comparison to the simulated ground truth when the sequencing data of only three gametes (b, c, d) are provided. For both panel A and panel B, the tested sequencing depths are 1X, 2X, 4X, 8X, 16X, 32X and 64X and the tested $Q_{net}$ cutoffs are 10, 20, 30, 40, 50, 60, 70, 80, 90, 100.

[24], which could be used in combination with RecombineX when processing sequencing data from randomly collected gametes.

For the scenario of fully viable tetrad, while the power of genotyping positively correlates with tetrad sequencing depth, a high accuracy is consistently maintained even with very limited tetrad sequencing depth. For example, the upper bound of RecombineX's raw and inferred genotyping error rates are estimated as $7 \times 10^{-4}$ and $2.64 \times 10^{-3}$ respectively for 1X-sequenced tetrad (Fig 4A and S2 Table). Aside from sequencing depth, $Q_{net}$ cutoff is another important parameter for RecombineX's genotyping analysis, which specifies the cumulative genotyping-supporting score leveraged over all mapped reads at a given marker site. A higher $Q_{net}$ cutoff helps to filter out ambiguous genotyping signals and thus ensures better genotyping accuracy. But it is also important to point out that the value of $Q_{net}$ roughly scales with sequencing depths, therefore setting it too high will compromise RecombineX's genotyping power for shallowly sequenced tetrads, leaving the genotypes of many markers as undetermined (Fig 4A and S2 Table). Luckily, since RecombineX employs a series of quality control filters (e.g., repetitive-region filter, CNV-region filter, genotyping purity filter, and biparental reciprocality filter) to minimize the chance of marker identification and genotyping errors, even with a

lenient $Q_{net}$ cutoff (e.g., 10), the chance of assigning false genotypes is still reasonably low. Therefore, in general, lenient $Q_{net}$ cutoffs such as 10 or 20 are recommended for tetrad with very shallow sequencing depth (e.g., 1X). In this case, the performance gain from getting more markers genotyped substantially outweighs the theoretical higher risk of getting erroneous genotypes, which is demonstrated by our simulation.

For the scenario of a partially viable tetrad, the error rates of our inferred gamete genotypes range from $6 \times 10^{-5}$ to $5.5 \times 10^{-3}$ across different sequencing depth and $Q_{net}$ cutoff combinations (Fig 4B and S3 Table). As demonstrated in our simulation, RecombineX can almost completely recover the true genotypes of the mimicked inviable gamete with 4X sequencing reads from the other three viable gametes (Figs 3D and 4B), which highlights the power of such missing genotype inference. In terms of application value, the inferred missing genotypes from the inviable gametes can be potentially used to map the genetic basis of gamete lethality. Moreover, such missing geno-type inference enables a more cost-effective design of trait-mapping experiments by making better use of shallowly sequenced samples. To better support these potential applications, RecombineX reports both raw and inferred genotyping results with rich graphical and textual outputs, making them highly amiable for further integration with genetic mapping tools such as R-qtl [25]. More-over, when all tetrads from the same processing batch come from the same cross, RecombineX can further summarize and plot the parental and missing allele frequencies along each chromo-some accordingly, which can facilitate discovering locus-specific allelic preference during genetic mapping analysis when batch size is reasonably large.

## Simulation-based validation for RecombineX's recombination profiling module

High quality genotyping performance of RecombineX lays the foundation for accurate recombi-nation event identification and classification. Here we assessed RecombineX's recombination pro-filing performance with the genotyping results obtained with different gamete sequencing depths and $Q_{net}$ cutoffs (Fig 5 and S4 Table). Our simulated tetrad has been introduced with 90 COs (each with an associated GC tract) and 65 NCOs, which translates into 90 Type 2 COs, 65 Type 1 GCs, and 90 Type 2 GCs (S5 and S6 Figs). It is worth mentioning that the power and accuracy of all genotype-based recombination analyses are ultimately linked to the density and distribution of available markers, which needs to be taken into consideration when interpreting the results of such analyses. For example, among our simulated recombination events, one CO event (which locates very close to the chromosome end), and five Type 2 GC events are inherently nondetect-able due to the lack of available markers in the corresponding genomic regions. We excluded these undetectable events from our downstream analyses, which left a total of 89 COs, 65 Type 1 GCs, and 85 Type 2 GCs to be identified by RecombineX.

Echoing with what we found for genotyping, the performance of recombination profiling positively correlates with tetrad sequencing depth. With moderate sequencing depth (e.g., depth > = 8X), RecombineX can recover all simulated CO and GC events, regardless of the specific $Q_{net}$ cutoff value used for genotyping. For more shallowly sequenced tetrads, the per-formance of RecombineX's recombination profiling began to be compromised due to the reduction of available markers with strong-enough genotype signals. This is more evident for GC events than for CO events due to the much smaller genomic footprint of GC tracts. Espe-cially, with high $Q_{net}$ cutoff but low sequencing depth, we observed some false positive events (referred as "incorrect events" in Fig 5) that were not simulated. These events were called due to the fact that some simulated event-associating markers were genotyped as "NA", leaving their nearby markers creating apparent event-supporting genotype patterns substantially devi-ated from the truth, which eventually leads to incorrect recombination event calls. Therefore,

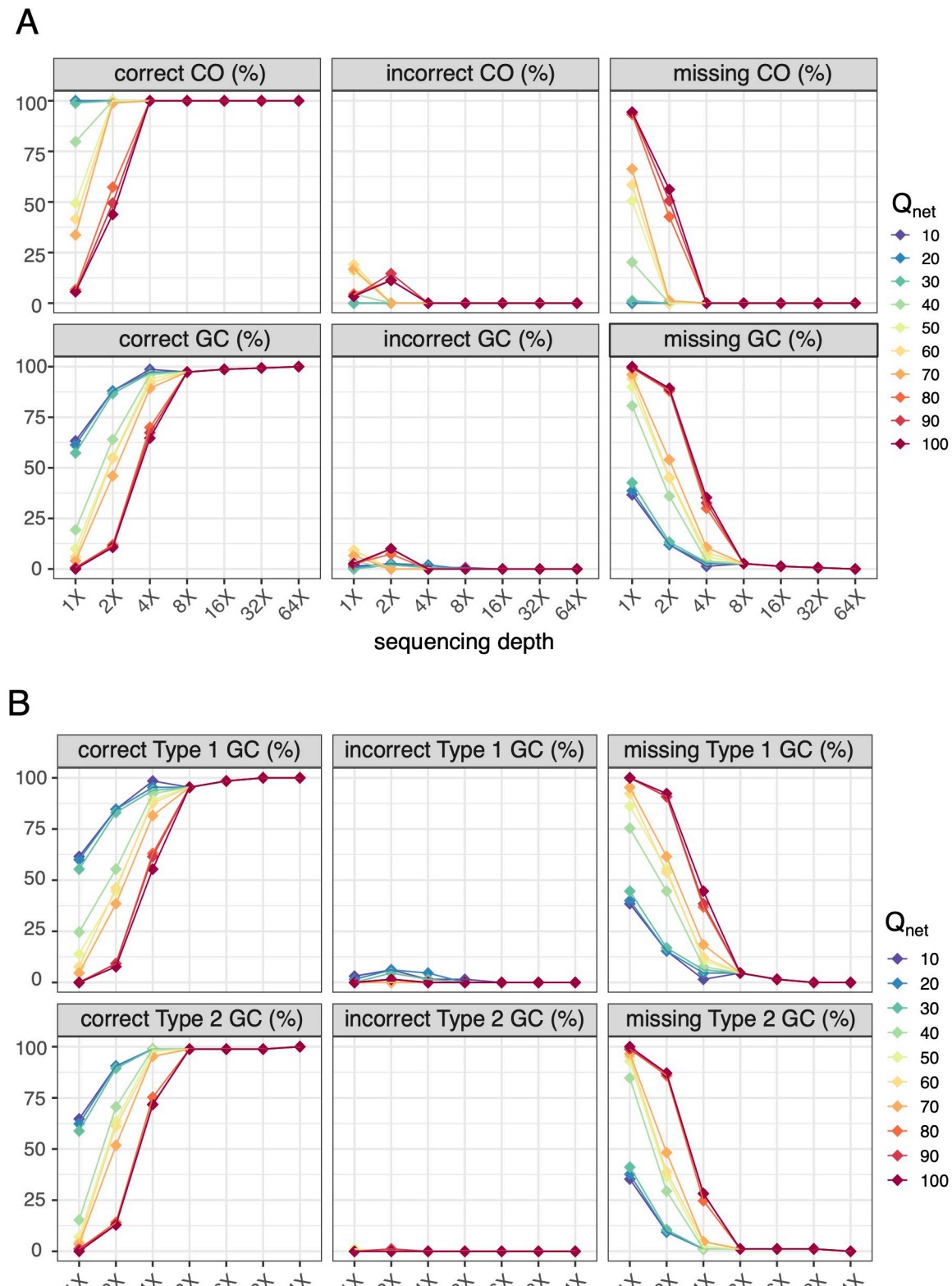

**Fig 5. Performance of recombination profiling with RecombineX based on simulated data.** The RecombineX-identified CO and GC events were compared with the simulated ground truth. (A) The percentage of correct, incorrect, and missing total CO and GC events identified by

RecombineX given different gamete sequencing coverage and $Q_{net}$ cutoffs. (B) The percentage of correct, incorrect, and missing Type 1 GC (NCO) and Type 2 GC (CO-associated GC) events identified by RecombineX given different gamete sequencing coverage and $Q_{net}$ cutoffs. Tested sequencing depth: 1X, 2X, 4X, 8X, 16X, 32X and 64X. Tested $Q_{net}$ cutoff: 10, 20, 30, 40, 50, 60, 70, 80, 90, 100.

lenient $Q_{net}$ cutoffs such as 10 or 20 are recommended for recombination profiling on shallowly sequenced tetrads, which helps to maintain a relatively high sensitivity of event calling without severe compromise in specificity.

## Applying RecombineX to real tetrad sequencing data

After systematically characterizing RecombineX's module-by-module performance with simulated data, we further applied it to real budding yeast (*Saccharomyces cerevisiae*) and green alga (*Chlamydomonas reinhardtii*) tetrads retrieved from previous studies for further demonstration [16,26]. For the budding yeast example, the sampled tetrads are derived from a cross between S288C and SK1 strains, for which the native genome assemblies of both parents are available [27]. Therefore, we performed RecombineX analysis in both reference-based and parent-based modes for this case. As for the green alga example, the sampled tetrads were derived from a cross between CC408 and CC2936 ecotypes, for which no native parental genome assembly is available. Therefore, we only executed RecombineX in the reference-based mode here. For both examples, we compared all recombination events automatically profiled by RecombineX against the curated recombination event lists reported by the respective original studies.

For the yeast example, 50,199 reference-based markers (mean intermarker distance = 234.35 bp) and 48,558 parent-based consensus markers (mean intermarker distance = 240.64 bp) were identified respectively (S5 Table). Our genotyping and recombination profiling analysis based on these markers shows a good concordance between RecombineX and the original study. In sum, RecombineX recovered 95.4% (378/396) of previous reported CO events and 89.9% (456/507) of previous reported GC events (Fig 6A and S6 Table). There are 3 CO and 21 GC events were only called by RecombineX, which were further verified by our manual inspection in Integrative Genomics Viewer (IGV) [28]. As for those only called by the original study (18 COs and 51 GCs), we found they were mostly filtered out by RecombineX at either marker identification or genotyping stage due to repeat/CNV-association or ambiguous genotypes. Therefore, the comparatively more events that were only identified by the original study are likely explained by the lack of explicit and stringent quality-control filters to account for repetitive regions, CNVs, and ambiguous genotypes during marker identification and gamete genotyping. For instance, by default, RecombineX requires a minimal genotype purity of 90%, meaning at least 90% of the mapped reads should support the same genotype signal at the corresponding marker position. By this standard, some of event-defining markers (and therefore the corresponding event) from the original study will be disregarded by RecombineX (S7 Fig). It is worth mentioning that while we found such a strong genotype purity filter is generally beneficial for preventing the inclusion of suspicious markers potentially derived from unreliable read mapping, users can still adjust this cutoff in RecombineX when needed (e.g., to study potential signatures of PMS).

For the green alga example, we identified 412,210 reference-based markers with an average intermarker distance of 260.28 bp (S5 Table). Comparing to the yeast example, we noticed a lower level of concordance of called recombination events between RecombineX and the original study for the green alga example, especially for GC events (Fig 6B and S7 Table). While RecombineX successfully recovered all CO events reported by the original study, it also

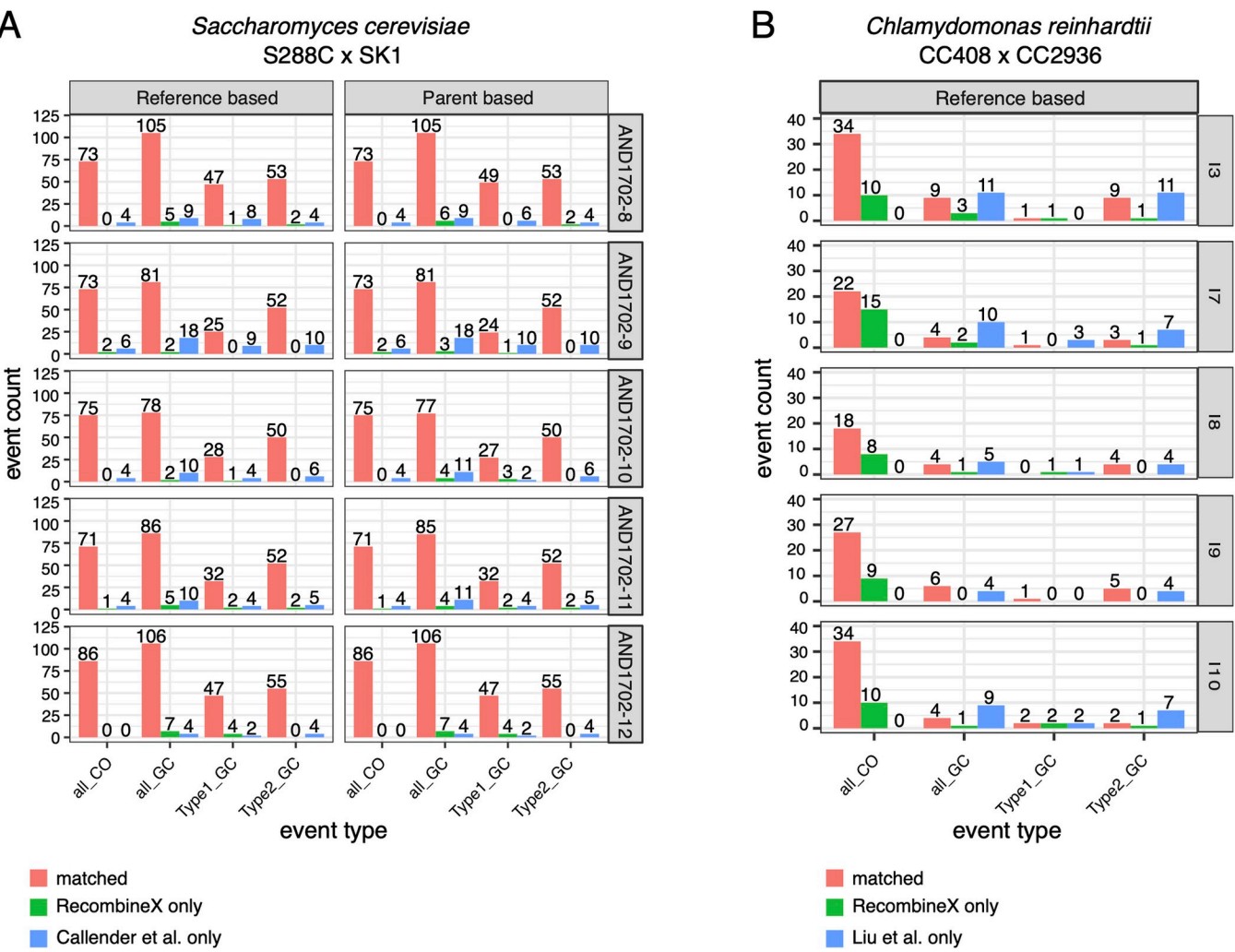

**Fig 6. Applying RecombineX to real yeast and green alga tetrads for recombination profiling.** Previously sequenced tetrads from yeast (AND1702-8, AND1702-9, AND1702-10, AND1702-11, AND1702-12) and green alga (I3, I7, I8, I9, I10) were used for re-processing with RecombineX, the results of which were further compared with the original studies (Callender et al. for the yeast example and Liu et al. for the green alga example). (A) The number of matched and unmatched total CO, total GC, Type 1 GC (NCO), and Type 2 GC (CO-associated GC) events identified by RecombineX (in both reference-based and parent-based modes) compared with the original study (Callender et al.) based on the yeast tetrad sequencing data. (B) The number of matched and unmatched total CO, total GC, Type 1 GC (NCO), and Type 2 GC (CO-associated GC) events identified by RecombineX (in reference-based mode) compared with the original study (Liu et al.) based on the green alga tetrad sequencing data.

identified a substantial number of CO events that were not reported before. By manually examining the local read mapping profiles of these events in IGV, we verified these RecombineX-only CO events as legitimate CO events (S8 Fig). We did notice that some of these events span over assembly gaps and potentially were filtered out in the original study for this reason. As for discrepant GC calls, our repetitive/CNV-region check and manual IGV inspection suggests that most of them are likely explained by the comparatively less stringent marker and genotype filtering of the original study, especially in complex genomic regions (S8 Fig). Also, It is worth noting that the size of GC tracts in green alga are substantially smaller (median size = 73 bp and 364 bp for Type 1 and Type 2 GCs respectively) when compared with yeast (median size = 1,681 bp and 1,841 bp for Type 1 and Type 2 GCs respectively)[16]. This means that the inclusion or exclusion of a single marker makes a big difference in GC event calling for green alga.

Last but not least, meiotic structural rearrangement could occur due to the numerous DSBs triggered during meiosis [29,30], which could significantly impact the genotypes of the affected gametes. RecombineX's bonus feature for CNV-profiling comes especially helpful in discovering such gamete-specific structural rearrangements. When analyzing the five yeast S288C-SK1 tetrads using RecombineX, we found two interesting cases of gamete-specific structural rearrangement that have not been noticed before (Figs 7 and 8). One such structural rearrangement is a large segmental duplication on chromosome IV (chrIV) of the gamete AND1702-8:a (i.e. the gamete a of tetrad AND1702-8) (Fig 7A). By design, RecombineX automatically flagged CNV regions like this and set the corresponding genotypes to "NA" as a conservative measure. To reveal the exact genomic arrangement of this duplication, we retrieved the monosporic isolate of this gamete and performed long-read-based genome sequencing and assembly. A joined comparison between the resulting *de novo* AND1702-8:a assembly and the S288C and SK1 genomes unraveled the intriguing nature of this gamete-specific rearrangement: a tandem duplication with the duplicated copies inherited from both parental backgrounds (Fig 7B and 7C). The breakpoints of this tandem duplication are associated with Spo11 DSB hotspots and TY-related repetitive sequences annotated along the S288C and SK1 genomes, which echoes similar observations made in mouse recently [31]. Comparatively, the rearrangement that RecombineX identified in the gamete AND1702-12:a (i.e. the gamete a of tetrad AND1702-12) appears more complex, in which both chromosome VI (chrVI) and chromosome IX (chrIX) are involved (Fig 8A). Here we also applied long-read sequencing and assembly to illuminate this complex rearrangement, which suggests both tandem and dispersed duplications have contributed to this complex rearrangement (Fig 8B). Again, parental genomic features such as Spo11 DSB hotspots and TY-related repetitive sequences are associated with the breakpoints, hinting their roles in triggering meiotic DSBs and driving gamete genome rearrangements (Fig 8B). These two cases of gamete-specific rearrangements demonstrated the power of resolving and understanding of tetrad formation and meiotic recombination within the native contexts of their parental genomes. As high-quality genome sequencing and assembly become increasingly affordable, future tetrad analyses are highly likely to shift away from the current reference-based convention and to embrace the parent-based new paradigm instead. In this sense, RecombineX with its built-in support for conducting analysis in parental genome space is expected to greatly facilitate such parent-based tetrad analysis.

In summary, we developed RecombineX as a generalized computational framework that automates the full workflow of marker identification, gamete genotyping, and tetrad-based recombination profiling in a high-throughput fashion, capable of processing hundreds of tetrads in a single batch. Aside from conventional reference-based analysis, RecombineX can also perform analysis based on parental genome assemblies, which enables analyzing meiotic recombination landscapes in their native genomic contexts. Additional features such as copy number variation profiling and missing genotype inference further extends its usage for various downstream analyses. Finally, RecombineX also ships with a dedicate module for simulating the genomes and reads of recombinant tetrads, which enables fine-tuned simulation-based hypothesis testing. This simulation module revealed the power and accuracy of RecombineX even when analyzing tetrads with very low sequencing depths (e.g., 1-2X). Tetrad sequencing data from the budding yeast *Saccharomyces cerevisiae* and green alga *Chlamydomonas reinhardtii* were further used to demonstrate the accuracy and robustness of RecombineX in tetrad analysis for organisms with both small and large genomes. As demonstrated in these examples, RecombineX unifies different functional modules under an integrated framework and provides a generalized one-stop solution for tetrad analysis. At the frontend, RecombineX shines in its modular design and parameter-rich customizability, making it highly amiable to different model systems and use scenarios. Behind the scenes, RecombineX implements thoughtful

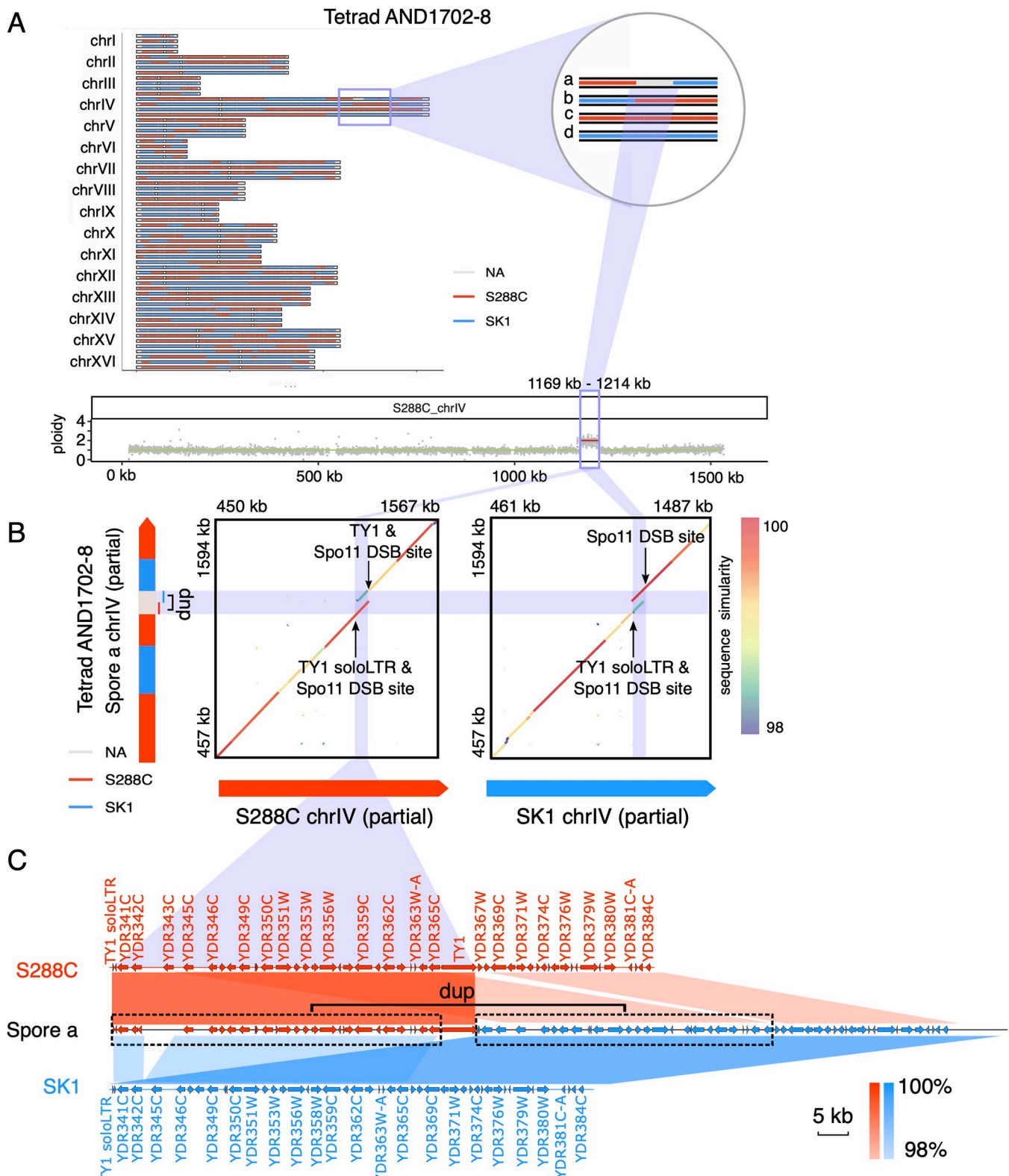

**Fig 7. RecombineX enables the discovery of complex structural rearrangements in the gamete AND1702-8:a.** (A) A 45-kb CNV identified by RecombineX in the gamete AND1702-8:a. (B) Genome sequence comparison between AND1702-8:a and its crossing parents (S288C and SK1) for the focal structural rearrangement and its flanking regions. Two genome comparison dotplots are shown for AND1702-8:a vs. S288C (left) and AND1702-8:a vs. SK1 (right) respectively, with their per-base sequence similarities depicted in rainbow colors. Previously annotated TY transposable element and Spo11 DSB hotspot

features are further indicated. The local genotype landscapes of the AND1702-8:a genome are shown on the left along the Y-axis, with the red and blue colors denoting the S288C and SK1 genotypes respectively. The duplicated region is colored in light grey and indicated with the "dup" label. (C) The gene synteny correspondence between AND1702-8:a and its crossing parents (S288C in red and SK1 in blue), with the red and blue shades representing different degrees of sequence similarity.

and rigorous algorithms, delivering trustable performance against biological and technical noises. The combination of these merits and the extended capacities of parent-based mode support, CNV profiling, missing genotype inference, batch processing, and tetrad simulation, together makes RecombineX a comprehensive platform for high-performance tetrad analysis. Especially considering that meiotic gamete genome sequencing from different natural and mutant backgrounds can now be acquired, RecombineX can greatly facilitate tetrad analysis across different genetic backgrounds. Finally, efforts have been made to perform tetrad analysis for mammals [32]. Such experiment, although remains technically challenging, might become conventional in near future along with the rapid advancement of single-cell sequencing. Therefore, we expect RecombineX to empowers future tetrad analysis for extended species.

## Materials and Methods

### Software Prerequisites

RecombineX is designed for a desktop or computing server running an x86-64-bit Linux operating system. Multithreaded processors are preferred to speed up the process since some time-consuming steps can be configured to use multiple threads in parallel. A stable internet connection is required for its installation. A number of standard Linux software compilation prerequisites are listed as below.

- bash (https://www.gnu.org/software/bash/)

- bzip2 and libbz2-dev (http://www.bzip.org/)

- gcc and g++ (https://gcc.gnu.org/)

- git (https://git-scm.com/)

- GNU make (https://www.gnu.org/software/make/)

- gzip (https://www.gnu.org/software/gzip/)

- libopenssl-devel

- libcurl-devel

- java runtime environment (JRE) v1.8.0 (https://www.java.com)

- perl v5.12 or newer (https://www.perl.org/)

- tar (https://www.gnu.org/software/tar/)

- unzip (http://infozip.sourceforge.net/UnZip.html)

- wget (https://www.gnu.org/software/wget/)

- zlib and zlib-devel (https://zlib.net/)

- xz and xz-devel (https://tukaani.org/xz/)

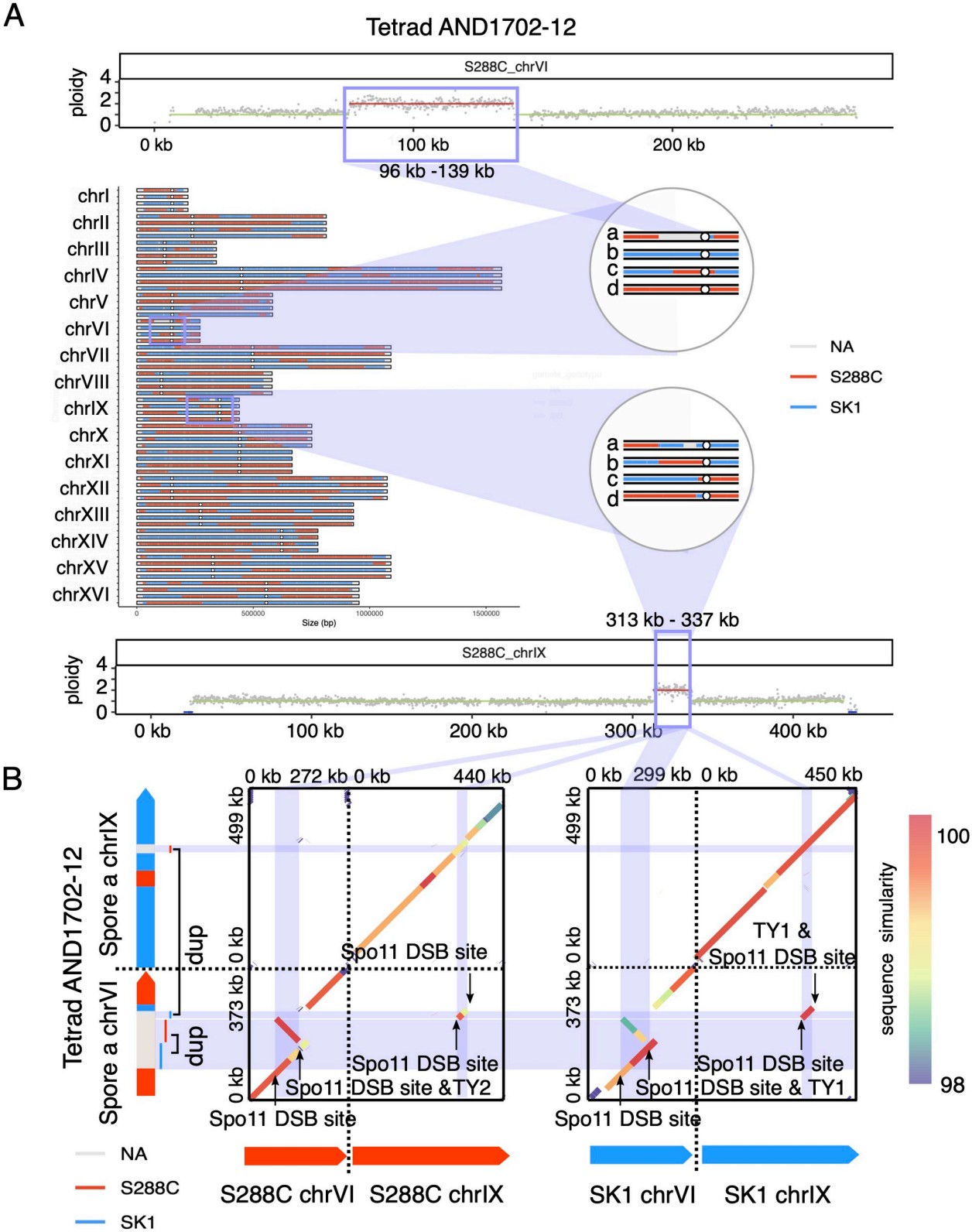

**Fig 8. RecombineX enables the discovery of complex structural rearrangements in the gamete AND1702-12:a.** (A) A more complex multi-chromosome-involved CNV identified by RecombineX in gamete AND1702-12:a. (B) Genome sequence comparison between AND1702-12:a and its crossing parents (S288C and SK1) for the focal structural rearrangement and its flanking regions. Two genome comparison dotplots are shown

for AND1702-12:a vs. S288C (left) and AND1702-12:a vs. SK1 (right) respectively, with their per-base sequence similarities depicted in rainbow colors. Previously annotated TY transposable element and Spo11 DSB hotspot features are further indicated. The local genotype landscapes of the AND1702-12:a genome are shown on the left along the Y-axis, with the red and blue colors denoting the S288C and SK1 genotypes respectively. The duplicated region is colored in light grey and indicated with the "dup" label.

### Software installation and configuration

In addition to the system-level prerequisites described above, RecombineX relies on a number of third-party bioinformatics tools for data analysis, all of which can be automatically installed and configured by RecombineX. A bash script is pre-shipped with RecombineX to perform such installation and configuration. A detailed list of these third-party tools and their underlying functions in RecombineX are provided (S8 Table).

### Expected input data

Depending on the available input data, RecombineX can be executed in two modes: 1) the reference-based mode and 2) the parent-based mode. The reference-based mode requires a reference genome assembly as well as the short reads of the two crossing parents. The parent-based mode requires the native genome assemblies of both crossing parents, while the short reads of the two crossing parents can be further used when available. Please note that the actual crossing parents can be either haploids or diploids, and in latter case, only homozygous SNVs will be considered for parental markers for downstream analysis. In both modes, short reads of individual gamete genome sequencing are further needed for gamete genotyping and recombination landscape profiling (if gametes are labeled based on their corresponding tetrad context).

### Software implementation

**Preprocessing for reference and parental genomes.** For both reference based and parent-based modes, RecombineX preprocesses the input genome(s) to generate necessary intermediate files for downstream analysis. These preprocessing steps include: cleaning up and relabeling the input genome file (by pre-shipped Perl scripts), indexing the input genome files by samtools [33] (version: 1.9; options: faidx), profiling repetitive sequences by windowmasker [34] (version: 1.0.0; options: -checkdup true -mk_counts), profiling GC content by bedtools [35] (version:2.27.1; options: makewindows -w 250), profiling mappability by gemtools [36] (version: 1.7.1; options: -m 0.02 -e 0.02).

**Reference-based parental marker identification.** The parental Illumina reads are processed with trimmomatic [37] (version: 0.38; options: PE -phred 33 ILLUMINACLIP:Tru-Seq3-PE-2.fa:2:30:10 SLIDINGWINDOW:5:20 MINLEN:36) to trim off adapter contamination and low quality bases. The trimmed reads are mapped to the preprocessed reference genome with bwa [38] (version: 0.7.17; option: mem). The resulting read mapping bam file is further filtered with samtools [33] (options: -q 30 -F 3340 -f 2) to only retain uniquely aligned and properly paired reads. Based on the filtered bam file, Picard Tools (https://broadinstitute.github.io/picard/) (version: 2.19.0) is used for sorting, mate information fixing, PCR duplicates removal, and indexing. Afterwards, GATK3 [39] (version: 3.6–6) is used for performing local read mapping around Indels for better accuracy. Based on the GATK3-realigned bam file, samtools is used again for mpileup file production (options: mpileup -C 0 -q 30), mapping depth calculation (options: depth -aa), and mapping summary statistics calculation (option: flagstat). After read mapping, FREEC [40] (version: 11.4) is used to perform sliding-window-based CNV profiling accordingly. Typically, FREEC requires a long list of

customized parameters tuned for the input genomes. In RecombineX, the suitable parameters are automatically estimated based on the preprocessed genome file. Freebayes [41] (version: 1.3.4) is used for SNV and Indel calling. The called variants are further processed by vt [42] (version: 0.57721) and vcflib [43] (version: 1.0.1) for variant decomposition (vt option: decompose_blocksub -a), normalization, annotation, and initial filtering (vcflib option: vcffilter -f QUAL > 30 & QUAL / AO > 1 & SAF > 0 & SAR > 0 & RPR > 1 & RPL > 1). Regardless of the ploidy of parental genomes, only homozygous SNVs are considered for downstream analysis. SNVs that fall in repetitive regions, CNV regions, or immediate Indel-flanking regions (10 bp) are further filtered out. As an additional CNV filtering, SNV sites with per-site mapping depths strongly deviating from the chromosome-wide median (e.g., > 1.5X chromosome-wide median or < 0.5X chromosome-wide median) are filtered out from the candidate marker list. The filtered SNVs derived from both parents are compared to each other to generate a candidate reference-based marker set comprising SNVs segregated between the two parents. Such candidate marker set is further leveraged with the mpileup file generated during read mapping to control for potential false negative and false positive from SNV calling. Finally, RecombineX pre-shipped with several plotting scripts in R to generate graphic reports on CNV and marker distribution along the reference genome coordinates.

**Parent-based parental marker identification.** Parent-based parental marker identification can be performed in two strategies: the genome-alignment-based marker identification and consensus-based marker identification. For the genome-alignment-based marker identification, RecombineX employs mummer3 [44] (version: 3.23) for genome alignment building (options: nucmer -g 90 -l 20 -c 65), filtering (options: delta-filter -1), and SNV extraction (options: show-snps -Clr). The extracted SNVs are further processed by vt [42] and vcflib [43] for decomposition, normalization, and annotation. Afterwards these marker candidates are filtered based on the repetitive profiling results generated at the parent-genome preprocessing step to remove repetitive-region-associated markers. Also, as the specific choice of query and target genome assemblies at the genome alignment step potentially could lead to directional bias, a reciprocal filter is further applied to only retain the reciprocal SNV calls recovered in both comparison directions (i.e., A to B and B to A). The corresponding filtered SNV set is defined as genome alignment based parental marker set. As for the consensus-based marker identification strategy, there are several extra steps to be taken. First, cross-parent read mapping is performed by mapping the Illumina reads of one parent to the genome assembly of the other parent. Accordingly, mapping based SNV, Indel, and CNV calling are carried out by freebayes [41] and FREEC [40]. Subsequently, the mapping based SNVs are further filtered by repetitive sequences and CNVs identified along the parental genomes. The detailed read mapping, variant calling, and variant filtering protocols are the same as those mentioned above, except for that both parental genome assemblies rather than a single reference genome is used in this case. Afterwards, the resulting mapping-based SNV calling sets are intersected with the genome-alignment-based SNV marker sets to derive a consensus marker set, upon which a final reciprocal filter is further applied to make sure the final consensus set is strictly symmetrical relative to the two crossing parents. For both genome-alignment-based and consensus-based marker set, RecombineX will also plot their respective marker distributions along the genome coordinates of both parental genomes.

**Reference-based gamete read mapping and genotyping.** RecombineX uses a strictly defined master sample table to document the metadata for each sequenced gamete sample, its Illumina reads, and its corresponding tetrad. According to this file, RecombineX will automatically perform reference-based read mapping and CNV profiling for each defined gamete sample based on the same protocols adopted for parental marker identification. As for genotyping, RecombineX takes the inputs from the mpileup file generated by gamete read mapping and

the reference-based parental marker list generated by marker identification to evaluate marker-specific reads support from each gamete across all marker sites. At each marker site, two quality control parameters are calculated. A $Q_{net}$ score is calculated as the cumulative sequencing score difference between the major allele (i.e., the best supported base) and all minor alleles (if any). In the meantime, a base purity score is calculated as the proportion of reads supporting the major allele at the corresponding marker site. A genotype is tentatively assigned only when both $Q_{net}$ and base purity meet their pre-defined cutoffs (50 for $Q_{net}$ and 0.9 for base purity by default). The tentatively assigned genotypes are further filtered based on the gamete specific CNV profiles generated by FREEC [40], during which the genotypes of markers falling in gamete specific CNV regions will be set as "NA". In addition to this raw genotyping result, RecombineX will generate another copy of genotyping result (labeled by the "inferred" tag in its file name) by further inferring the possible missing genotypes based on a general 2:2 parental allele segregation ratio across the tetrad. For both raw and inferred genotyping results, RecombineX will make both tetrad-based and batch-based genotyping plots with pre-shipped R scripts to visualize the composition and segregation of two parental genetic backgrounds across the genome.

**Parent-based gamete read mapping and genotyping.** The protocol used by RecombineX for parent-based gamete read mapping and genotyping is largely the same as that used for reference-based gamete read mapping and genotyping, except for now the analyses are separately performed based on both parental genome assemblies. Therefore, after reads mapping, two genotyping results will be obtained separately, each based on the genome space of a single parent. These two genotyping calls will be cross validated with each other. Those markers with conflicted genotyping calls will be ignored for downstream analysis (i.e., their genotypes will be reassigned to "NA"). Like reference-based analysis, features such as gamete CNV profiling and missing genotype inference are fully supported for parent-based analysis.

**Reference- and parent-based recombination profiling.** The recombination profiling procedures in the reference-based and parent-based modes are essentially the same, except that the parent-based mode will perform the analysis twice, each based on the genome space of a single parent. RecombineX implemented a modified version of the original recombination event profiling algorithms used by ReCombine [19] to cover all foreseeable scenarios (S5 and S6 Figs). Briefly, RecombineX scans through the tetrad-wide genotypes at every marker site to classify them into different categories based on the segregation ratio of the parental alleles (e.g., 2:2 or 3:1 or 1:3 or 4:0 or 0:4). Consecutive markers with identical segregation ratios are grouped together, based on which preliminary linkage blocks are identified. By ignoring the remaining markers with missing genotypes, the calculated preliminary linkage blocks are further extended to form final linkage blocks. The outer bounds of each final linkage block are defined by the midpoint of the outermost markers of this linkage block and their immediate flanking markers. Users can restrict such linkage block identification operation by modifying the minimal number of supporting markers (default: 1 marker) and minimal block size (default: 1 bp) parameters. According to the identified final linkage blocks, RecombineX systematically examines the genotype switch patterns and the number of gametes involved in genotype switches between each pair of adjacent linkage blocks to classify the local recombination events. Those recombination events that are in close adjacency (controlled by the "merging range" parameter) can be further merged when needed. Upon the completion of the calculation, detailed tabular reports on the marker-wide parental allele segregation ratio, preliminary and final linkage blocks, as well as lists of recombination events will be reported.

**Recombinant tetrad genome and reads simulation.** RecombineX performs recombinant tetrad genome simulation based on an input genome assembly and a list of clearly defined

parental markers. Both reference assembly and native parental genome assembly can be used as the input here, as long as the accompanying parental marker list is based on the same genome coordinates. With these inputs, RecombineX first simulate the two parental genomes by projecting parental markers to the input genome assembly. Based on the simulated parental genomes, CO and GC events are further simulated based on various user-specified parameters, which include the number of CO and GC events, the ratio of CO and GC events, the mean and standard deviation of GC tracts, etc. These recombination events are randomly placed into the four resulting gametes. For both simulated parental and gamete genomes, paired-end Illumina reads are further simulated by ART [45] with user-specified sequencing depth.

## Simulation based analysis

**Genome and reads simulation for parental marker identification.**   To evaluate the performance of parental marker identification, a pair of hypothetical crossing parents, P1 and P2, were simulated for this study. The genome of P1 is an exact copy of the budding yeast *S. cerevisiae* reference genome (version: R64-2-1_20150113) retrieved from *Saccharomyces* Genome Database (SGD) with the mitochondrial genome excluded. Based on the same reference genome, the genome of P2 was further generated by simuG [46] (GitHub commit version: 212ea1f) in a two-pass manner to randomly introduce 60,000 SNVs and 6,000 Indels (options: -refseq SGDref.genome.fa -snp_count 60000 -titv_ratio 2.0 -indel_count 6000 -seed 20190518 -prefix yeast_60kSNP_6kINDEL) as well as six CNVs (options: -refseq yeast_60kSNP_6kIN-DEL.simseq.genome.fa -cnv_count 6 -cnv_gain_loss_ratio 1 -duplication_tandem_disperse-d_ratio 1 -cnv_max_copy_number 4 -centromere_gff SGDref.centromere.gff). The centromere annotation of the *S. cerevisiae* reference genome (distributed with the reference genome) was used for the second pass to prevent the simulated CNVs from surpassing centromeres.

For the simulated genome of P1 and P2, 150-bp paired-end Illumina reads were further simulated by ART [45] (version: MountRainier-2016-06-05; options: "-p -l 150—qprof1 HiSeq2500L150 –qprof2 HiSeq2500L150 -f <depth> -m 500 -s 10 -na -rs 20210210). Here we simulated a wide range of sequencing depths (10X, 20X, . . ., 100X) for exploring the influence of sequencing depth on parental marker identification. The simulated parental genome and reads for P1 and P2 were fed into RecombineX for parental marker identification. The identified markers following both reference-based and parent-based protocols were compared with the initially simulated SNVs between P1 and P2.

**Recombinant tetrad simulation for gamete genotyping and recombination profiling.** The aforementioned P1 genome and the consensus markers that RecombineX identified based on 100X parental reads were used as the inputs for tetrad genome simulation. A total of 90 COs and 65 GCs were simulated with their size distribution parameters determined based on real yeast tetrads: mean Type 1 GC size = 2250 bp, standard deviation of Type 1 GC size = 2200, mean Type 2 GC size = 2500 bp, standard deviation of Type 2 GC size = 2000 bp, min GC size = 100 bp, max GC size = 5000 bp [9]. Random seed was set to 20210210 for this simulation. The introduced recombination events and the resulting genotypes were used as ground truth sets for downstream comparison. For each simulated gamete genome, paired-end Illumina reads were further generated by ART with varied depths (1X, 2X, 4X, 8X, 16X, 32X, 64X). The simulated gamete reads were processed with RecombineX in both reference-based and parent-based modes. The generated gamete genotyping and recombination profiling results were compared with the ground truth generated during our simulation. The impacts of different $Q_{net}$ cutoffs (e.g., 10, 20, 30, . . ., 100) were thoroughly explored during this process.

## Real tetrad-based analyses

Two real tetrad sequencing datasets retrieved from previous studies [16,26] were used to run the full workflow of RecombineX. The first dataset includes five tetrads derived from the budding yeast *S. cerevisiae* cross S288C x SK1, for which both reference-based and parent-based analyses were performed (S9 and S10 Tables). The SGD yeast reference genome (version: R64-2-1_20150113) was used for the reference-based analysis, while our previously generated long-read-based genome assemblies for S288C and SK1 [27] were used for the parent-based analysis. The second dataset includes five tetrads derived from the green alga *C. reinhardtii* cross: CC408 x CC2936 (S9 and S10 Tables), for which only reference-based analysis was performed. For this analysis, we retrieved the alga *C. reinhardtii* (v5.5) reference genome from Ensembl Plants (https://plants.ensembl.org). For both yeast and green alga datasets, a $Q_{net}$ cutoff of 50 was used and no adjacent recombination event merging was applied. The RecombineX profiled recombination events were compared with the events reported in the original studies, which were retrieved via the following links respectively:

yeast tetrads: http://dx.doi.org/10.5061/dryad.g6s2k

green alga tetrads: https://figshare.com/s/a95156f0ed5272b9109e

In our comparison, two events were considered "match" only if they completely agreed with each other in event types, genomic locations, and involved gametes. For events that were identified by RecombineX or the original studies alone, we further examined the gamete read alignment in the IGV browser [28] (version: 2.8.13) to understand the specific causes of such disagreement.

## Oxford Nanopore sequencing of yeast gametes AND1702-8:a and AND1702-12:a

To take a closer look at the structural rearrangements identified by RecombineX for yeast gamete AND1702-8:a and AND1702-12:a, we retrieved the corresponding strains stocked in Dr. Alain Nicolas's lab at Institut Curie (Paris, France). Upon receiving the yeast cells, we grew them in 10–15 ml YPD (2% peptone, 1% yeast extract, 2% glucose) at 30˚C for overnight (220 rpm). A total number of cells less than $7 \times 10^9$ were used for DNA extraction. High molecular weight (HMW) DNA was extracted by the QIAGEN Genomic-tip 100/g kit according to the "QIAGEN Genomic DNA handbook" for Yeast. DNA quantity and length were controlled by the Qubit dsDNA HS Assay and the Nanodrop 2000 Spectrophotometers respectively. Library preparation and ONT sequencing were performed based on the protocol of "1D Native barcoding genomic DNA with EXP-NBD104 and SQK-LSK108" obtained from Oxford Nanopore Technologies Community. The FLO-MIN106 MinION flow cell was used for sequencing.

## Genome assembly, annotation, and comparison for yeast gametes AND1702-8:a and AND1702-12:a

The nanopore reads were processed with our previously developed LRSDAY pipeline [47] (version: 1.6.0) for *de novo* genome assembly and comprehensive feature annotation. The internal protocols employed by LRSDAY are briefly described as follows. The raw nanopore-sequencing fast5 reads are processed with Guppy (version: 3.2.4) for basecalling and demultiplexing. The resulting fastq reads are further trimmed with Porechop (version: 0.2.4; options: —discard_middle) and filtered with Filtlong (version: 0.2.0; options:—min_length 1000—mean_q_weight 10—target_bases 750000000). The filtered reads are assembled with Canu (version: 1.8; options: -s genomeSize = 12.5m -nanopore-raw). The raw Canu-assembly is further polished with both nanopore (sequenced in this study) and Illumina reads (retrieved from

the original study). Three successive rounds of long-read-based polishing are performed by Racon [48] (version: 1.4.7) and Medaka (https://github.com/nanoporetech/medaka) (version: 0.8.1; options: -m r941_flip235). Another three successive rounds of short-read-based polishing are performed by Pilon [49] (version:1.23;—fix snps,indels). The polished assembly is further processed with Ragout [50] (version: 2.2) and circulator [51] (version: 1.5.5; option: fixstart—genes_fa ATP6.cds.fa—min_id 90) for reference-based scaffolding and mitochondrial assembly improvement. The resulting final nuclear and mitochondrial genome assemblies are further annotated by Maker3 [52] (version: 3.00.0-beta) and Mfannot (https://github.com/BFL-lab/Mfannot) (version: 1.35) respectively, with additional reference-based gene orthology identification performed by Proteinortho [53] (version: 5.16b). Other important genomic features such as centromere, tRNA, Ty transposable elements, core-X elements, Y' elements were also annotated by dedicated modules implemented in LRSDAY. The fully assembled and annotated genomes of the gametes AND1702-8:a and AND1702-12:a obtained in this way were further compared with the native genome assembly of their crossing parents (S288C and SK1) by Mummer3 [44], BLAST+ [54] (version: 2.2.31+; options: -blastn) and Easyfig [55] (version: 2.2.3; options: -i 98 -min_length 1000 -filter -f1 T -f2 1000) regarding both sequence similarity and annotated genomic features. The Spo11 DSB hotspot annotation used in such comparison was retrieved from the literature [56].

## Supporting information

**S1 Fig. The modular workflow design of RecombineX.** RecombineX consists of seventeen task-specific modules, with six modules dedicated for the reference-based mode (colored in yellow) and eight modules dedicated for the parent-based mode (colored in blue). As for the three remaining modules, two are designed for both reference-based and parent-based modes (colored in green), with the last one for simulation analysis (colored in red).
(TIF)

**S2 Fig. Overview of the RecombineX directory system.** The pre-shipped top-level directories and individual files of RecombineX are denoted with solid lines. Additional directories and files to be generated during the installation of RecombineX are denoted with dashed lines.
(TIF)

**S3 Fig. Overview of the RecombineX parental marker identification algorithms.** Two marker identification modes are supported: the reference-based mode (colored in yellow) and the parent-based mode (colored in orange).
(TIF)

**S4 Fig. Overview of the RecombineX gamete genotyping algorithms.** Two genotyping modes are supported: the reference-based mode (colored in yellow) and the parent-based mode (colored in orange).
(TIF)

**S5 Fig. Overview of the RecombineX recombination event classification scheme.** The definition and example of different CO and GC types are shown in panel a and panel b respectively. This classification scheme is designed based on the original ReCombine recombination event classification scheme [19] with additional modifications.
(TIF)

**S6 Fig. Overview of the RecombineX recombination event identification algorithm.** This algorithm is designed based on the original ReCombine algorithm [19] with additional

modifications.
(TIF)

**S7 Fig. Manual examination of unmatched recombination events from the yeast S288C-SK1 tetrads in IGV.** The read alignments of parent and gamete reads are visualized in IGV with event-defining SNP markers shown in colors.
(TIF)

**S8 Fig. Manual examination of unmatched recombination events from the green alga CC408- CC2936 tetrads in IGV.** The read alignments of parent and gamete reads are visualized in IGV with event-defining SNP markers shown in colors.
(TIF)

**S1 Table. RecombineX's performance in parental marker identification performance based on simulated data.**
(XLSX)

**S2 Table. RecombineX's gamete genotyping performance based on simulated tetrads with all four gametes.**
(XLSX)

**S3 Table. RecombineX's gamete genotyping performance based on simulated tetrads with only three viable gametes.**
(XLSX)

**S4 Table. RecombineX's recombination event profiling performance based on the simulated tetrad.**
(XLSX)

**S5 Table. Summary of parental marker identification for real cross examples with RecombineX.**
(XLSX)

**S6 Table. RecombineX's recombination event profiling performance based on real yeast tetrads.**
(XLSX)

**S7 Table. RecombineX's recombination event profiling performance based on real green alga tetrads.**
(XLSX)

**S8 Table. Description of third-party software packages that will be automatically downloaded and installed during RecombineX's installation.**
(XLSX)

**S9 Table. Yeast and green alga tetrad sequencing datasets employed in this study.**
(XLSX)

**S10 Table. Yeast and green alga parental genome sequencing datasets employed in this study.**
(XLSX)

## Acknowledgments

We thank Dr. Johan Hallin (Gothenburg University, Gothenburg, Sweden) and Ms. Jessica Chevallier (CRCM, Marseille, France) for beta testing and valuable discussion. We thank Dr.

Alain Nicolas (Institut Curie, Paris, France) and Ms. Sophie Loeillet (Institut Curie, Paris, France) for locating and sharing the original gamete samples of AND1702-8:a and AND1702-12:a.

## Author Contributions

**Conceptualization:** Gianni Liti, Jia-Xing Yue.

**Data curation:** Jing Li, Jia-Xing Yue.

**Formal analysis:** Jing Li, Jia-Xing Yue.

**Funding acquisition:** Jing Li, Bertrand Llorente, Gianni Liti, Jia-Xing Yue.

**Investigation:** Jing Li, Jia-Xing Yue.

**Methodology:** Jing Li, Bertrand Llorente, Gianni Liti, Jia-Xing Yue.

**Project administration:** Gianni Liti, Jia-Xing Yue.

**Resources:** Bertrand Llorente, Gianni Liti, Jia-Xing Yue.

**Software:** Jia-Xing Yue.

**Supervision:** Gianni Liti, Jia-Xing Yue.

**Validation:** Jing Li, Bertrand Llorente, Gianni Liti, Jia-Xing Yue.

**Visualization:** Jing Li, Jia-Xing Yue.

**Writing – original draft:** Jing Li, Jia-Xing Yue.

**Writing – review & editing:** Jing Li, Bertrand Llorente, Gianni Liti, Jia-Xing Yue.

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
