## [Decision Letter · Decision Letter 0]

14 Mar 2022

Dear Dr Yue,

Thank you very much for submitting your Methods entitled 'RecombineX: a generalized computational framework for automatic high-throughput gamete genotyping and tetrad-based recombination analysis' to PLOS Genetics.

The manuscript was fully evaluated at the editorial level and by independent peer reviewers. The reviewers appreciated the attention to an important topic but identified some concerns that we ask you address in a revised manuscript

We therefore ask you to modify the manuscript according to the review recommendations. Your revisions should address the specific points made by each reviewer.

[LINK]

Yours sincerely,

Sarah Zanders

Guest Editor

PLOS Genetics

Chengqi YI

Section Editor: Methods

PLOS Genetics

Your work has been reviewed by three experts and they all agree it is a method that would be useful to readers of PLOS Genetics after minor revisions. Importantly, the reviewers noted difficulties in running the program themselves, which limits the usefulness of the method. In addition, the reviewers offer useful suggestions to improve the clarity of the manuscript and figures.

Reviewer's Responses to Questions

**Comments to the Authors:**

Reviewer #1: This is an interesting methods paper describing a program called RecombineX that can be used to perform tetrad analysis from whole genome sequencing data. There are a number of interesting features, including a module for inferring the genotype of a missing 4th member of tetrad that is only represented by three sequences.

The program is admirably already publicly available. I was able to install RecombineX with some minor issues. There were multiple dependencies required for installation that were not installed in the "install_dependencies.sh" install script. For the most part these were easy to manually install and restart the script. However, the script was also unable to install DNAcopy on my system or to detect my manual installation of DNAcopy despite a reasonable amount of effort on my end- I had to edit out the command in the install script in order to complete installation.

Once that was done I was able to follow the manual to run the sample data using parental sequence data (modules 00-05). These performed as expected (except for a typo in the manual described below).

Generally, I have two main complaints about the application:

1) Use of multiple sequential modules that are not linked by a single pipeline script

2) Running the modules requires manually editing individual script files

I would encourage the authors to write a pipeline script with reasonable default values that allows a user to run the program from the command line interface without substantial manual building of files. This would lower the barrier to others using RecombineX in their own work.

All that said, the paper itself is in pretty good shape and does a pretty satisfying job of demonstrating the capabilities of RecombineX, including identifying CNVs and structural changes missed by previous approaches and validating them with nanopore sequencing. I would encourage the authors to make several of their figures much larger. They are difficult to read as is.

Specific Comments:

Figure 1 "Gamete Reads" part in upper right is a little misleading. This suggests every paired end read is a fusion of parent 1 and parent 2. Pretty sure this isn't as intended.

Page 4, Wang et al. 2012 is double cited "Wang et al. 2012, 2012."

Figure 4 A-D are unreadably small on the page. By zooming in, you can also see there are some labels that are partially covered in white boxes.

Figure 6 is very small.

Figure 7 again is unreadably small.

Error in the manual for the sample data:

cd ./../ RecombineX.03.Gamete_Read_Mapping_to_Reference_Genome.sh

bash RecombineX.03.Gamete_Read_Mapping_to_Reference_Genome.sh

The RecombineX. should be deleted from the first of those commands.

Reviewer #2: The authors developed a software package to analyze meiotic recombination along the genome from sequencing data of the four meiotic products (tetrad analysis). The pipeline can be employed to perform tetrad analysis in any organism in which it is possible to sequence the genomes of the four products resulting from a single meiosis. The package includes several modules that allow performing all the different steps of the analysis required for tetrad analysis when starting from raw genomic sequencing reads: genotyping of the parents to identify polymorphic markers, genotyping the meiotic products and identification of the recombination events from the genotypes. Marker identification can be performed based on a reference genome of the species in question or directly from sequencing data of the specific parental lines used for the cross. The software also includes a module to simulate the genotypes of the meiotic products that can be used to test deviations from empirical data. After testing the package analyzing simulated genotypes, the authors assess its performance with tetrad data from budding yeast and a green alga, showing that the resulting recombination profiles are very similar to those determined in the original works. As the authors put it, the main advantages of the software are that all the steps needed for tetrad analysis can be performed using a single package, and the fact that it can be employed for any organism where tetrad analysis can be performed.

Below are my specific comments ordered from mayor to minor.

1. Can the package deal with diploid parental genomes? Several strains and organisms have been sequenced in their diploid state and their genome assemblies are not phased (PMID: 35210580, for example). It would be important to discuss the potential/limitations of the software in that respect.

2. The two modes for identifying markers could be better explained. For example, in the parent-based mode it is not clear whether genome assemblies of the parents are a prerequisite or whether only having sequencing reads of the parents is enough. This is also confusing when the authors perform the analysis in the green alga since they mentioned that they only used the reference mode given that no parental assembly was available. In the reference mode the reference assembly is needed and reads of one parent, or the assembly of one parent? Please clarify the requirements.

3. Given that the package needs different software to be run it would be important to mention the computational requisites in the methods so that the feasibility of running the package is clear.

4. When mentioning that almost all previously identified events in the yeast example were recovered it may be better to give exact percentage or number of recovered events. Similarly when mentioning events only found by RecombineX, and throughout, when comparisons with previous studies are done. For example, in Figure 6 it seems there are more events only identified by Callender et al., it would be better to include the actual numbers in the text. It may be also important to mention very briefly how previous events were identified to understand the possible reasons for the discrepancies.

5. Figure 4E and F it is not clear why lower Qnet cutoffs would give higher number of correct genotypes.

6. The author mentioned that it is possible that changing the default parameters of RecombineX it may be possible to increase the concordance with past results. Have they actually attempted trying different parameters with the real yeast and alga data?

7. In Figure 5 it would be important to clarify what are incorrect events (middle panels). Especially because the tendencies in the graphs of correct and missing events seem to be exact reciprocal of each other, leaving no room for incorrect events.

8. In Figure 4, the genotypes of tetrads in panels A to D are too small, it is not possible to see the exact genotypes. Either enlarge the figure or only show some of the chromosomes, but larger so that the genotypes can be distinguished. In panels E and F, 10 Qnet cutoffs were tested, but only a couple of lines can be seen in the figure, always less than 10. Also, from the color legend of the Qnet cutoffs it would seem that the cutoffs tested are a continuous, but they are rather 10 discrete cutoffs (same in Figure 5).

9. Figure 7 has the same problem as the upper panels of Figure 4, the tetrad genotypes are too small to see the recombination events. In general, in this figure there is a lot of information in a small space making it hard to understand.

10. In Figure 3 it is not clear what is meant by “markers whole genome alignment” and why the line is steady unlike the reference- and parent-based markers.

11. I am not sure Figure 2 should be a main figure since it is a rather technical representation of the software that does not bring much to the reader unless they are direct users of the package.

12. Apart from the organisms that the authors mention where tetrad analysis can be performed, there have also been efforts to do them in animals (PMID: 25151354). Although these experiments are currently technically very challenging, they may become more conventional in a near future. It may be worth mentioning them as a possible future use of the software.

13. Would be good to add reference PMID: 21481229 when referring to PMS.

14. The reference to IGV should be used the first time IGV is mentioned and “IGV browser” may be clearer, at least for the first instance, than just IGV.

15. Some of the colors of lines in Figure 3 are difficult to distinguish (blue, purple and green).

16. In the sentence “Moreover, additional tools are available for reconstructing the gamete-tetrad correspondence map from random gametes (Sakhanenko et al. 2019),…” it would be important to mention that these are previously developed tools. The start of the sentence misleads the reader to think that the tools are part of RecombineX.

17. In the legend of figure 1B instead of saying “Tetrad genotyping” it may be better to say “Gamete genotyping” as it is used in the figure itself.

18. In the sentence “…the sampled tetrads are derived from a cross between S288C and YPS128 strains,…” it may be better to use SK1 instead of YPS128 as it is used in the rest of the article.

19. Check typos and syntax in the following sentences:

• have substantially expand our

• inspecting individual meiosis event, which

• generation of computational solution for high-throughput

• for polymorphic markers identification, gametes genotyping

• Upon the obtaining of parental markers

• making this assumption holds in general.

• inference, RecombineX’s raw genotyping function do not require any tetrad information as the priori when performing raw genotyping.

• original study, with RecombineX that completely recovered

• Type 1 and Type 2 GC respectively

• which could make significant impacts on the genotypes

Reviewer #3: Li et al describe RecombineX, a software package for processing and analyzing genetic recombination data from tetrad analysis. The manuscript is clear, well written, and the software seems useful. It is significantly more powerful than the ReCombine package from a different group (Anderson et al 2011, cited) which, as far as I know, is the only comparable piece of software. In particular, RecombineX can be run in a mode that does not require one of the two parents to be the “reference” strain of the species (this is a substantial limitation of the older ReCombine package). Li et al also tested RecombineX extensively by simulation, allowing them to conclude that it performs very well and that it can even detect most recombination events when sequencing coverage of the gametes is very low, about 1-2x.

RecombineX will be a useful tool. It might be argued that the manuscript would be a better fit to PLOS Computational Biology, but in my view the audience for this software is geneticists, so the manuscript is appropriate for PLOS Genetics.

My main comment is that I ran into difficulties when I tried to install RecombineX on my computer (Macbook Pro laptop), so I was unable to actually run the program myself. I will attach some parts of the log file as an appendix to this review in the hope that the authors can troubleshoot it. This is not a major criticism. The authors have provided extensive documentation on GitHub to support the software, and I know that it is extremely difficult to anticipate all the installation problems and software dependencies that users will encounter. In my case, the installation script stopped running during the section “Installing Perl modules” because installation of the Math::Random library failed. When I commented out this line in the install_dependencies.sh script, I then got multiple errors during the attempted installation of BedTools. In the end, no env.sh file was created so I was not able to run RecombineX at all.

I would like to suggest: (1) The authors should try troubleshooting the installation on a few different computer systems. (2) At the moment it looks like RecombineX will not run at all unless every other program that it depends on is installed successfully and correctly. Consider providing an option for users to do a minimal installation, in which RecombineX could start running even if some of its functions won’t work. For example, I can see that it installs the NCBI BLAST package, but I think that this is only required for the non-reference genome mode, so it should be possible to run RecombineX in reference-genome mode even if BLAST is not installed. It would also be useful to provide documentation explaining which pieces of external software are required for each function of RecombineX, so that a user could install only the pieces of external software that they will actually need.

The structural rearrangement in Figure 7C is unusual and interesting, and I think that it deserves to be mentioned in the Abstract.

Page 5, end of Introduction: Please include a description of what input data is required by RecombineX. I can see that it requires Illumina reads files from 4 gametes in a tetrad, and either a reference genome assembly or assemblies of each of the 2 parents. Does it also require reads files from the 2 parents? Are any other input files required?

Figure 4: In panels A-D, it is quite difficult to see the red and blue colors because the gray boxes around each chromosome are quite thick. I can see the colors on my screen if I zoom-in a lot on the PDF, but on a printout and onscreen at 100% magnification they are too difficult to see. This comment also applies to Fig 7A,B. Consider making the chromosome boxes thinner, or use a single line instead of a box.

For the S. cerevisiae tetrads, was the second parent in the cross strain YPS128 (page 15) or SK1 (pages 17 and 24)? And do the tetrad names begin with AND7 (Fig 6 legend), or AND17 (text on page 17) ??

(See also attached PDF file of installation errors).

**Have all data underlying the figures and results presented in the manuscript been provided?**

Reviewer #1: Yes

Reviewer #2: Yes

Reviewer #3: Yes

PLOS authors have the option to publish the peer review history of their article (what does this mean?). If published, this will include your full peer review and any attached files.

Reviewer #1: No

Reviewer #2: No

Reviewer #3: No

---

## [Editor Report · Decision Letter 1]

14 Apr 2022

Dear Dr Yue,

We are pleased to inform you that your manuscript entitled "RecombineX: a generalized computational framework for automatic high-throughput gamete genotyping and tetrad-based recombination analysis" has been editorially accepted for publication in PLOS Genetics. Congratulations!

Yours sincerely,

Sarah Zanders

Guest Editor

PLOS Genetics

Chengqi YI

Section Editor: Methods

PLOS Genetics

Comments from the reviewers (if applicable):

The authors have addressed the reviewers' concerns and improved the paper. This tool will be useful for folks studying meiotic recombination. I have a few minor suggestions the authors could address prior to final submission:

In the description of meiosis in the intro, i suggest changing 'cell division' to 'nuclear division' at several spots as the first meiotic division does not always generate distinct cells.

The text says: “RecombineX that automates the full workflow of tetrad analysis based on any organisms and genetic backgrounds.“ I suggest suggest changing to ‘based on any organism or genetic background.’

The text says: “it lacks the power and resolution for analyzing individual meiosis event.” I suggest “analyzing an individual meiosis” or “individual meioses.”

The text says: “Notably, the improvement of the consensus marker count become marginal with sequencing depth > 50X.” I suggest 'becomes.'

suggest becomes

The text says: “For the scenario of partially viable tetrad.” I suggest 'of a partially viable tetrad.'

**Data Deposition**

http://datadryad.org/submit?journalID=pgenetics&manu=PGENETICS-D-22-00080R1

**Press Queries**

---

## [Editor Report · Acceptance letter]

2 May 2022

PGENETICS-D-22-00080R1 

RecombineX: a generalized computational framework for automatic high-throughput gamete genotyping and tetrad-based recombination analysis 

Dear Dr Yue, 

We are pleased to inform you that your manuscript entitled "RecombineX: a generalized computational framework for automatic high-throughput gamete genotyping and tetrad-based recombination analysis" has been formally accepted for publication in PLOS Genetics! Your manuscript is now with our production department and you will be notified of the publication date in due course.

With kind regards,

Livia Horvath

PLOS Genetics

On behalf of:
